# What Influences Health Professionals’ Recommendations for Non-Scheduled Childhood Vaccinations? A Qualitative Study of Health Professionals’ Perspectives in Three Provinces of China

**DOI:** 10.3390/vaccines9121433

**Published:** 2021-12-03

**Authors:** Jiejie Cheng, Shiyu Lin, Chaoqi Wu, Natasha Howard, Jiatong Zou, Fiona Yueqian Sun, Mei Sun, Tracey Chantler

**Affiliations:** 1Department of Health Policy and Management, School of Public Health, Fudan University, 130 Dong’an Road, Shanghai 200032, China; 18211020050@fudan.edu.cn (J.C.); 20211020128@fudan.edu.cn (S.L.); 17301020145@fudan.edu.cn (C.W.); 2NHC Key Laboratory of Health Technology Assessment, Fudan University, 130 Dong’an Road, Shanghai 200032, China; 3Department of Global Health and Development, London School of Hygiene & Tropical Medicine, 15-17 Tavistock Place, London WC1H 9SH, UK; natasha.howard@nus.edu.sg (N.H.); Tracey.Chantler@lshtm.ac.uk (T.C.); 4Saw Swee Hock School of Public Health, National University of Singapore and National University Health System, 12 Science Drive 2, Singapore 117549, Singapore; 5Shanghai Municipal Center for Disease Control and Prevention, Department of Organization and Personnel, 1380 West Zhongshan Road, Shanghai 200336, China; zoujiatong@scdc.sh.cn; 6Department of Infectious Disease Epidemiology, London School of Hygiene & Tropical Medicine, University of London, Keppel Street, London WC1E 7HT, UK; Fiona.Sun@lshtm.ac.uk

**Keywords:** vaccines, recommendations, economic incentives, motivation, qualitative research

## Abstract

Recommendations by health professionals are important for vaccines that are not included in national schedules. This study explored health professionals’ perspectives on recommending non-scheduled (user-fee) childhood vaccinations in China, identifying key influences on professionals’ interactions with caregivers. We conducted individual semi-structured interviews with 20 health professionals from three provinces in China and analyzed data thematically using deductive and inductive coding. Health professionals from all three provinces were uncomfortable about being perceived to encourage parents to accept vaccines that incurred a fee. They provided information about non-scheduled vaccines but emphasized parental autonomy in decision-making. Rural parents were less aware of unscheduled vaccines and health professionals were more likely to encourage parents living in more affluent areas to consider these vaccines; varicella vaccine was preferred by parents as a way of preventing school absence. Economic incentives for unscheduled vaccines were given to staff at most study sites, although the amount given varied widely. These variations meant that staff receiving lower incentives were not motivated to promote non-scheduled vaccines if their workload was high; on the contrary, those receiving higher incentives were more likely to promote these vaccines. Health professionals need more guidance on how to recommend unscheduled vaccines in an informative, positive and appropriate manner. It is evident that parents’ awareness of these vaccines, and their economic circumstances, influence vaccinators recommendation practice. Economic incentives prompted health professionals to recommend non-scheduled vaccines; however, the application of such staff incentives varied widely in China. To adopt appropriate economic incentives, professional organizations should develop protocols for the use of incentives that account for their influence on recommendation practices. Suitable recommendation policy needs to balance basic salaries with performance-based incentives, consider overall workload, and include monitoring and evaluation of economic incentives.

## 1. Background

Recommendations by Health Professionals (HPs) are important catalysts for parental uptake of vaccines, especially for those not included in routine national immunization schedules [1]. The Chinese public health system delivers two complementary immunization programs for children under 6 years of age. The first (Category I) is the routine childhood Expanded Program of Immunization (EPI) [2,3], which is a government funded program that includes vaccines that are required according to the law on vaccine administration in China. The second (Category II) consists of supplementary optional vaccines that incur a fee for service-users. Category I and II vaccines are delivered at points of vaccination (POVs), found at a range of health facilities including township and community health centers. HPs based at POVs are responsible for offering vaccination services and promoting the uptake of vaccines that can prevent common diseases. During their daily work they are responsible for providing vaccine information, addressing service users’ questions and administering vaccines. Previous research has identified a variety of factors that influence HPs’ vaccination recommendation behaviors: individual characteristics (e.g., educational level [4,5], working years [4,6,7,8,9], age [10]), knowledge of vaccines [5,8,11], vaccine confidence [12,13], and motivation related to income or economic incentives [5,7,14,15,16].

Economic incentives have been shown to influence HPs’ vaccine recommendation behaviors in a variety of geographic contexts. A study conducted by Fairbrother et al. in the United Kingdom (UK) in 1999 detected a sharp increase in immunization rates when HPs received cash bonuses [17]. Similarly, Watkins found that the introduction of incentive payments for healthcare workers delivering the UK national influenza vaccine program resulted in a significant increase in uptake in recommended vaccines [14]. In China, research indicates that HPs’ willingness to recommend non-EPI vaccines is related to incentive payments [8,15,18]. These quantitative studies suggest that additional income HPs can receive from administering non-EPI vaccines increases their willingness to recommend these vaccines. This evidence describes a link between monetary incentives and HPs’ motivation to deliver immunization programs, which merits further investigation. For example, how does the receipt of economic incentives affect HPs’ interactions with service-users. Does it make a difference if the services on offer to service-users are free or require vaccine recipients to pay a fee? The analysis presented in this paper contributes insights into these questions.

This analysis draws on data from a research project that examined caregivers of children under the age of 6 (i.e., parents, and guardians) and HPs’ vaccination behaviors in three provinces in China. Preliminary findings from interviews with HPs indicated that they had reservations about recommending non-EPI vaccines. To examine this in more detail we asked HPs to elucidate underlying reasons for this hesitation. For the purpose of this analysis, we define economic incentives as financial incentives that increase an HP’s income, including cash bonus or direct payment.

## 2. Methods

### 2.1. Study Design

We applied a qualitative study design, informed by a critical realist perspective as described by Maxwell [19] that involved conducting cross-sectional semi-structured interviews with HPs who deliver or manage vaccination services. Critical realist approaches enable pragmatic engagement with ‘real world’ public health, to which concepts and theories refer.

### 2.2. Research Questions

(i).What are HPs working in China perspectives and experiences of recommending non-EPI vaccines?(ii).What influences interactions with parents and caregivers for HPs in China?(iii).What is the relationship between economic incentives and HPs working in China recommendation behaviors for vaccines not included in the national routine childhood immunization schedule?

### 2.3. Study Setting

We applied purposive sampling to select the study sites [20]. Firstly, we selected three provinces across Eastern, Central, and Western China to represent regional socioeconomic disparities that can influence public health system capacity to deliver immunization services. Secondly, we selected 1 rural county and 1 urban district per province to account for China’s dichotomous rural–urban service delivery structure. Finally, we selected 3-5 POVs from each district and ended up with a sample of 20 POVs from rural and urban areas across three provinces.

### 2.4. Participant Recruitment

We recruited one EPI manager, or one senior immunization HP, at each of the 20 POVs (Table 1). To be eligible to participate, HPs needed to be familiar with non-EPI vaccines and responsible for delivering immunization services. We applied purposive sampling to obtain a range of perspectives and achieve maximum variation in terms of gender, age and professional experience [20].

### 2.5. Data Collection

We interviewed 20 people (i.e., 1 per POV) in private locations chosen by interviewees from April to July 2019. Interviews were conducted face to face in Mandarin by one of 5 trained interviewers. We used topic guides pre-tested in Shanghai (i.e., at 1 rural and 1 urban public facility, 2 corresponding district/county Chinese Center for Disease Control and Prevention (CDC) offices, and the Shanghai CDC) to facilitate the interviews. The guides included space to record essential socio-demographic information (e.g., gender, age, educational background, years of service) and open-ended questions about the provision and uptake of EPI and non-EPI vaccines at POV, vaccination workload (EPI and non-EPI), attitudes towards vaccination (EPI and non-EPI), recommendation behaviors for non-EPI vaccines, caregivers’ vaccination awareness (especially for non-EPI vaccines), and incentives (especially for non-EPI vaccines). Interviewees provided informed consent before they were interviewed and received a token (e.g., baggage tag) to thank them for their time.

### 2.6. Data Analysis

The interviews were audio-recorded and transcribed verbatim by the research team. The data were analyzed thematically in NVivo software (Version 8, QSR international) using a coding framework developed from the topic guides and researchers’ observations and impressions [21]. We generated initial codes during the familiarization with the data stage, which we then reviewed and collated as defined themes and sub-themes before mapping these systematically to the data set

## 3. Results

### 3.1. Participant Characteristics

Our sample included 20 HPs: 7 from Henan, 6 from Sichuan, and 7 from Guangdong province. Most were women (12/20), nurses (8/12), aged 35–40 (11/20), working in rural areas (12/20), educated to junior college level or below (9/20), who had worked in immunization for >10 years (12/20) (see Table 1).

### 3.2. Analytical Themes

Our analysis generated three themes: (i) negative connotations of ‘recommending’ non-EPI vaccines; (ii) influences of location and workload on non-EPI vaccine recommendations; and (iii) influence of economic incentives on recommendations.

### 3.3. Negative Connotations of ‘Recommending’ Non-EPI Vaccines

Most HPs in all three provinces reported telling caregivers about non-EPI vaccines to fulfil their responsibilities of informing caregivers and allowing them to decide whether to get their child vaccinated. Their general perspectives were that vaccines could prevent infectious diseases, but they did not want to be perceived as actively recommending non-EPI vaccines. Only two interviewees, both from rural areas in Henan province, said they did not recommend non-EPI vaccines because they considered EPI vaccines to be sufficient for disease prevention and did not want to increase the economic burden on caregivers.

Almost every health professional was uncomfortable about recommending non-EPI vaccines and did not want to be seen as actively recommending anything; they just wanted to ensure caregivers were informed of the options. They emphasized that this was notification not recommendation and they did not want to be seen as making money by selling vaccines.

“That’s not recommendation, it’s just notification. It’s our job to clearly tell parents, we must tell, the information about non-EPI vaccine, such as the pros and cons of non-EPI vaccine, what disease it prevents, etc. But we cannot say that you have to choose non-EPI vaccines. It’s their own choice to get their children vaccinated non-EPI vaccines.” P12, Sichuan province.

### 3.4. Influences of Location and Workload on Non-EPI Vaccine Recommendations

In all three provinces, urban health workers reported greater likelihood of recommending non-EPI vaccines. Rural HPs reported caregivers’ vaccination awareness as low and suggested rural caregivers would feel uncomfortable being recommended non-EPI vaccines because they might connect this with the action of HPs seeking extra benefits. To avoid this misunderstanding, rural HPs were less inclined to recommend non-EPI vaccines.

Across the three provinces, we found that recommendation behaviors for non-EPI vaccines differed depending on participant locations. HPs in Henan province said it was not easy for them to recommend non-EPI vaccines, with most indicating their biggest obstacle was the lack of caregiver knowledge and awareness about vaccination. Caregivers in Henan were not sufficiently familiar with the differences between EPI and non-EPI vaccines, making it difficult for HPs to interest them in non-EPI vaccines.

“Most people consider that vaccines are free. When you recommend non-EPI vaccines to them, they can’t understand the difference between EPI and non-EPI vaccines and can’t accept these self-paid vaccines. It will cause financial burden for caregivers and also misunderstanding for ourselves, so I don’t recommend non-EPI vaccines unless caregivers take initiate to ask for them.”P04, Henan province.

Only three non-EPI vaccines were routinely provided by POVs in Henan province (i.e., Varicella, Hand, Foot and Mouth, Influenza). Some HPs indicated that caregivers were more familiar with these vaccines so recommendations were easier. For example, many caregivers knew that once their children were infected by varicella at school, they could be absent from school for several weeks, potentially affecting their academic performance. Therefore, varicella vaccination was described as popular in Henan province, with up to half of caregivers purchasing varicella vaccines.

“The uptake of varicella vaccine is higher. Parents’ knowledge of varicella vaccine is more sufficient, and they are more concerned about it, which is easier for us to recommend.”P03, Henan province.

Guangdong caregivers’ vaccination knowledge was more comprehensive than in Henan, potentially due to higher socioeconomic development in the province.

“Nowadays most caregivers’ knowledge is very rich and their awareness of vaccination is very high, sometimes I think they are more professional than me.”P20, Guangdong province.

Guangdong caregivers were reported to be relatively familiar with non-EPI vaccines and could afford to buy these vaccines. This made it easier for HPs to routinely recommend non-EPI vaccines, although they indicated that their willingness to recommend non-EPI vaccines decreased when they were very busy. Many participants stated that their workload was heavy and during peak vaccination seasons they had to extend their working hours and thus tended to avoid recommending non-EPI vaccines to reduce their workload.

“During the peak season, we are busy with vaccinating from morning to noon. Our centre is supposed to be closed at 12:00 p.m., but sometimes we even can’t have lunch until 1:00 p.m. because we have to get the children who have already come to be vaccinated. The waiting area was crowded and noisy. It is very uncomfortable. Sometimes when I am too busy, I just give them the consent forms and tell them to read it by themselves. To tell the truth, I don’t want more people come because we are too tired.”P18, Guangdong province.

The situation for HPs in Sichuan was in between that reported in Henan and Guangdong provinces. Their workload was relatively lighter than in Guangdong and caregiver awareness of different vaccines was higher than in Henan. Hence HPs in Sichuan voiced the strongest willingness to recommend non-EPI vaccines and even described taking the initiative to do so.

“The non-EPI vaccine is voluntary and self-paid. I recommend it and caregivers choose to get their children vaccinated by themselves. I think caregivers in our area have strong compliance and the uptake of non-EPI vaccines is quite high. Most people generally accept non-EPI vaccines...our publicity about vaccination starts from the newborn. When the vaccination card is established and the vaccination starts, I start to introduce the pneumonia vaccine, the five-unit vaccine, and so on. Now, young parents are more than happy to pay for non-EPI vaccines.”P13, Sichuan Province.

“We have six doctors here, so we are not particularly busy. We always inform them of the pros and cons of the non-EPI vaccines regardless whether these vaccines are in stock. We also tell them what kind of disease these vaccines can prevent... Our non-EPI vaccination volume can reach half of the total vaccination volume”P05, Sichuan Province.

### 3.5. Influences of Economic Incentives on Non-EPI Vaccine Recommendations

Most (15/20) POVs implemented performance-based incentive systems to encourage HPs to recommend and administer non-EPI vaccinations. Participants were very open about this and described varying financial bonuses that we have categorized as: high (i.e., 5–14 Yuan per injection), medium (i.e., 3–4 Yuan per injection), and low or no incentives (i.e., 0–1 Yuan) with 1 Yuan equivalent to approximately USD 0.15 in 2021.

Participants were reluctant to mention the relationship between workload, monetary value of incentives and their motivation to recommend non-EPI vaccines. However, by interviewing HPs based at different POVs, we were able to identify that HPs were more likely to promote non-EPI vaccination when the monetary value of the incentive was attractive enough to outweigh concerns about workload. In Henan province, the monetary value of incentives per non-EPI vaccine was two to three times more than that of Sichuan, and higher than that of Guangdong. In practice, even if the vaccination awareness of caregivers in Henan province is relatively low, the uptake of non-EPI vaccines, such as varicella vaccine was not low.

“Take the varicella vaccine recommendation for example, when it’s time for varicella vaccine, we tell caregivers varicella vaccine is preventable. And we will explain how much it costs. Then we will call caregivers make the appointment for varicella vaccine. Nearly half or more parents will get their children vaccinated after recommendation.”P03, Henan province (High 8 Yuan bonus).

HPs receiving lower economic incentives were less likely to recommend non-EPI vaccines when they were busy. This was particularly notable in Guangdong province. During peak vaccination season (e.g., for influenza or the beginning of school terms) low bonuses did not motivate HPs to extend their work hours.

“It’s not much money if there are extra non-EPI services and I am very busy. It’s unnecessary to recommend non-EPI vaccines for that money. We must spend time focusing on EPI vaccination because it’s related to government assessment for Category I vaccines coverage.”P11, Sichuan province (Low 2 Yuan bonus).

“We can only get 0.7 yuan per non-EPI injection, but we are so tired. I would rather leave work on time than work overtime for such a small amount of money.”P18, Guangdong province (Low 0.70 Yuan bonus).

HPs who received no economic incentives were unlikely to recommend non-EPI vaccines no matter how strong caregivers’ vaccination awareness was, or how light their workload was. Several indicated that they did not actively recommend non-EPI vaccines because they did not get additional money for providing non-EPI vaccination. This disincentivized them especially when facing caregivers who showed little interest or could not afford non-EPI vaccines.

“I don’t recommend non-EPI vaccines. There is no difference from the aspect of income whether I recommend non-EPI vaccines and provide more non-EPI vaccination or not. Sometimes there are parents who come to consult me about vaccination and are willing to pay for it, then I will tell them the information about non-EPI vaccines.”P01, Henan province (no bonus).

“I only recommend EPI vaccines. I can’t get income from non-EPI vaccination. It doesn’t matter for me whether I provide more non-EPI vaccination or not. If parents ask for non-EPI vaccines, I would provide non-EPI services.”P02, Henan province (no bonus).

## 4. Discussion

Our analysis focused on factors that influence HPs’ recommendations of non-EPI vaccines. We found that HPs’ recommendations for these non-scheduled vaccines were primarily influenced by economic incentives and busy working conditions.

Most HPs we interviewed were uncomfortable about the topic of non-EPI vaccine recommendation, despite providing caregivers with information about non-EPI vaccines as part of their routine work. They did not want their recommendation to be misunderstood as profit-seeking. Previous studies suggest that endorsement by professional organizations and perceived social norms influence health professionals’ recommendation behaviors. Mays et al. [22] found endorsement by a professional organization enhanced nurses’ willingness to recommend vaccination, because they felt that recommending immunization was a normative behavior. Similarly, Hswen et al. [23] found HPV vaccination endorsement by professional organizations was positively associated with physicians’ knowledge and recommendation practices. Thus, to encourage ‘recommendation-friendly’ working environments and enhance parents’ confidence, government or professional organizations should develop information and communication guidelines for health professionals to support consistent and transparent recommendation practice for non-EPI vaccinations. These guidelines would need to provide clear explanations of why some vaccines are free and some not, and who receives payment for those that are not free.

HPs’ recommendation willingness was influenced by perceived caregiver awareness of vaccination, with lower knowledge reducing motivation. This supports previous studies indicating that the willingness of patients or caregivers to receive vaccines influenced health professional confidence in recommending vaccination [13]. A study in South Africa [24] found nurses were more likely to recommend HPV vaccination when they felt adolescents were willing to accept it. To raise awareness of non-EPI vaccines health professionals could strengthen health education for caregivers during quiet periods in the clinic or via online portals. This may help caregivers understand the value of non-EPI vaccines, especially if health professionals commence vaccination education during pregnancy.

HPs’ recommendation behaviors were significantly influenced by their workload, with staff being less likely to recommend non-EPI vaccines if this increased their workload. This supports previous studies [7,15] that highlighted that staff who reported that non-EPI vaccination increased their workload were less likely to recommend these vaccines. One way of addressing this limitation could be for HPs to focus on non-EPI vaccines during quiet periods and POVs could also try to reallocate staff to provide additional cover during busy periods.

HPs’ recommendation behaviors were also notably influenced by economic incentives. Recommending non-EPI vaccination improved the income of most health professionals we interviewed. It was evident that incentives were a positive influence on HPs’ recommendation behaviors, which aligns with the literature [8,15]. What this study adds is that the positive influence of economic incentives on recommendation willingness can remain consistent even when caregiver awareness is not high. As a type of performance-based incentive, economic incentives have been effective in improving uptake and delivery of health services [25,26,27,28,29]. While the reported responses to incentives appear promising, our findings indicate that the financial value of incentives is important. Additional concerns about implementing economic incentives from the literature include neglect of non-incentivized tasks and distorted motivation among health professionals [30]. Thus, to use economic incentives effectively, it is important to account for the potential problems they may cause. In this case, appropriate income distribution must be established to avoid excessive and unnecessary non-EPI provision for economic benefit. This distribution system should balance basic salary and performance-based incentives (e.g., avoiding overly low basic salary and overly high incentives), which may distort motivation [31]. Governments should consider adopting recommendation motivation policies that balance basic salary, performance-based incentives, and overall workload. Moreover, it is essential to monitor and evaluate intended and unintended outcomes of interventions involving incentives [32].

### Limitations

This study has several limitations. First, since we only included three provinces, findings cannot be generalized to all provinces or districts in China. Second, while we included a range of HPs who deliver vaccines on the frontline, we did not interview other key stakeholders responsible for managing immunization services at district, regional or national level (e.g., public health staff working for the Chinese Center for Disease Control and Prevention), who would likely have provided different and additional perspectives. Thirdly, since this was a qualitative study aimed at understanding practice, we did not capture data about the use of economic incentives for promoting non-EPI vaccines in a larger sample of POVs and different geographical areas, which would be remedied in the follow-up studies.

## 5. Conclusions

Most HPs routinely provided information about non-EPI vaccines, but their recommendations were influenced by perceived caregiver awareness of vaccination, work intensity, and economic incentives. A more recommendation-friendly atmosphere, with appropriate and consistent government guidelines for recommendation practices and the use of incentives, could help health professionals adjust the timing of non-EPI vaccine recommendations, and conduct necessary health education during quieter periods. Suitable recommendation policy needs to balance basic salaries with performance-based incentives, consider overall workload, and include monitoring and evaluation of economic incentives.

## Figures and Tables

**Table 1 vaccines-09-01433-t001:** Characteristics of the research participants.

Provider Characteristics	Henan Province *n* (%)	Sichuan Province *n* (%)	Guangdong Province *n* (%)	Total
Gender				
Male	5 (62.5)	0 (0)	3 (37.5)	8
Female	2 (16.7)	6 (50)	4 (33.3)	12
District/county level				
Rural	5 (45.4)	3 (27.3)	3 (27.3)	11
Urban	2 (22.2)	3 (33.3)	4 (44.5)	9
Age (years)				
<35	0 (0)	1 (50)	1 (50)	2
35–45	3 (27.3)	5 (45.5)	3 (27.3)	11
>45	4 (57.1)	0 (0)	3 (42.9)	7
Education				
Junior college or below	3 (33.3)	2 (22.2)	4 (44.5)	9
Undergraduate	1 (20)	1 (20)	3 (60)	5
Missing	3 (50)	3 (50)	0 (0)	6
Major				
Clinical medicine	5 (100)	0 (0)	0 (0)	5
Nursing	1 (12.5)	4 (50)	3 (37.5)	8
Preventive medicine	0 (0)	0 (0)	3 (100)	3
Others	1 (100)	0 (0)	0 (0)	1
Missing	0 (0)	2 (100)	0 (0)	2
Working years in immunization				
<10	2 (50)	1 (25)	1 (25)	4
10–20	2 (33.3)	2 (33.3)	2 (33.3)	6
>20	2 (33.3)	0 (0)	4 (66.6)	6
Missing	1 (25)	3 (75)	0 (0)	4

## Data Availability

The data presented in this study are available on request from the corresponding author.

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
