# Peer review of "What Influences Health Professionals’ Recommendations for Non-Scheduled Childhood Vaccinations? A Qualitative Study of Health Professionals’ Perspectives in Three Provinces of China"

_vaccines, 2021, doi:10.3390/vaccines9121433_

Round 1
Reviewer 1 Report
I sincerely congratulate the authors because the topic is exciting and relevant for public health. The manuscript is well written. I feel that the qualitative methodology is suitable for the scientific question. However, there are some minor issues to clarify. I will explain all of them in the following comments:
C1: The introduction is short, but I feel that it goes clear to the main point. Anyone can predict the objective of the study. The introduction is well written.
C2: Minor comments for the methods section. Please, explain the method the authors used to select the districts randomly. In some cases, depending on the technique used, the authors can introduce bias.
C3: Following C1, please, describe how the authors selected the sample. Inside each district, the 3-5 participants were randomly selected from a population of health professionals who met the inclusion criteria, or was it a convenience sample?
C4: The results section is also well written. Congratulations, because the manuscript is very interesting and hooks you to finish it.
C5: The discussion section is also well written. However, I feel that the limitations section is underdeveloped. All the conclusions that the authors deduce from the results are correct, but as the study design is qualitative, classical limiations of qualitative analyses may bias all the results. Please, develop this section more, providing information about which quantitative methodology should be used to test the hypotheses derived from the study. In line with C2-3, please, include a limitation about the sample selection if it was not entirely random.
Author Response
Response to the comments by Mei Sun et al.
Dear Editors,
We thank you and the reviewers for your kind words and valuable comments, which we have used to revise our manuscript. Our point-by-point responses are listed below with reference to where and how we have made changes to the manuscript.
Reviewer 1
Point 1:The introduction is short, but I feel that it goes clear to the main point. Anyone can predict the objective of the study. The introduction is well written.
Response: Thank you for your kind words.
Point 2:Minor comments for the methods section. Please, explain the method the authors used to select the districts randomly. In some cases, depending on the technique used, the authors can introduce bias.
Response: We have sought to provide more detail on our sampling strategy in the methods section. As is customary in qualitative research we used a purposive (not random) sampling strategy to ensure that our study sites were sufficiently diverse in terms of socio-demographic/economic characteristics. In addition, the local CDC staff helped us to select a wider range of POVs on the consideration of more information but not better performance rates. (line 102-106 p3)
Point 3: Following C2, please, describe how the authors selected the sample. Inside each district, the 3-5 participants were randomly selected from a population of health professionals who met the inclusion criteria, or was it a convenience sample?
Response: Following the answer for C2, we selected Points of Vaccination (where health professionals involved in immunization work) purposively. We have sought to provide more detail on our participant recruitment strategy in the methods section. The inclusion criteria for participants was that they had to be health professionals, who were familiar with non-EPI vaccines, and were willing to and able to provide consent to participate in an interview.(line 110-114 p3)
Point 4:The results section is also well written. Congratulations, because the manuscript is very interesting and hooks you to finish it.
Response: Thank you for your encouraging feedback.
Point 5:The discussion section is also well written. However, I feel that the limitations section is underdeveloped. All the conclusions that the authors deduce from the results are correct, but as the study design is qualitative, classical limitations of qualitative analyses may bias all the results. Please, develop this section more, providing information about which quantitative methodology should be used to test the hypotheses derived from the study. In line with C2-3, please, include a limitation about the sample selection if it was not entirely random.
Response: Thank you. We agree that there are limitations associated with the purposive sampling approach and the qualitative research design. In response, we have expanded this section of the manuscript. (line 327-336 p8)
Reviewer 2 Report
I have read this paper with interest
It is not yet sufficiently clear that the focus is on pediatric vaccinations, this should be reflected in the title and the abstract.
I miss the reflection on the fact that if vaccination practices result in individual benefits to the healthcare provider, that this may also affect the confidence of parents and the public for the scheduled vaccinations ? This is even more the case, as category 1 and 2 vaccines are provided at the same locations.
Is the ‘advice to give for category 2 vaccines standardized, as e.g. the summary provided on varicella vaccination is rather limited (like the likelihood of early adulthood primary infection).
We need more information on the structure and organization of such a vaccination location to better understand the setting (and are vaccinations of category 1 also ‘free’ of choice, or ‘obligated’ (legal status?) We need information on ‘similar’ systems on pediatric vaccinations in the world (specific on funding related to ‘category 2’ vaccinations (not clear if the incentives relate ?
Author Response
Response to the comments by Mei Sun et al.
Dear Editors,
We thank you and the reviewers for your kind words and valuable comments, which we have used to revise our manuscript. Our point-by-point responses are listed below with reference to where and how we have made changes to the manuscript.
-Reviewer 2
Point 1: I have read this paper with interest. It is not yet sufficiently clear that the focus is on pediatric vaccinations, this should be reflected in the title and the abstract.
Response: Thank you for this helpful comment. The keywords ‘childhood’ and ‘children’ have been added in both the title and the abstract. (line 3, 4,24 p1)
Point 2: I miss the reflection on the fact that if vaccination practices result in individual benefits to the healthcare provider, that this may also affect the confidence of parents and the public for the scheduled vaccinations? This is even more the case, as category 1 and 2 vaccines are provided at the same locations.
Response: This is an important point that we have sought to expand on in the discussion section:“Thus, to effectively use economic incentives to encourage health professional recommendations, we must consider the potential problems they may cause. Appropriate income distribution must be established to avoid excessive and unnecessary non-EPI pro-vision for economic benefit. This distribution system should balance basic salary and performance-based incentives (e.g. avoiding overly low basic salary and overly high incentives, which may distort motivation[31]. Governments should consider adopting an appropriate motivation policy that balances basic salary, performance-based incentives, and overall workload. Moreover, strong monitoring and evaluation of any incentives programme are needed.”(line 317-325 p7-8)
Point 3: Is the ‘advice to give for category 2 vaccines standardized, as e.g. the summary provided on varicella vaccination is rather limited (like the likelihood of early adulthood primary infection).
Response: The advice for administering and recommending category 2 vaccination needs to be standardized. We have thought about this and added the following in the discussion section. “Thus, to encourage ‘recommendation-friendly’ working environments and enhance parents’ confidence, government or professional organizations should develop communication guidelines for health professionals to support consistent and transparent recommendation practice for non-EPI vaccinations. These guidelines would need to provide clear explanations of why some vaccines are free and some not and who receives payment for those that are not free.” (line 284-289, p7)
Point 4: We need more information on the structure and organization of such a vaccination location to better understand the setting (and are vaccinations of category 1 also ‘free’ of choice, or ‘obligated’ (legal status?) We need information on ‘similar’ systems on pediatric vaccinations in the world (specific on funding related to ‘category 2’ vaccinations (not clear if the incentives relate ?
Response: Thank you for your suggestion. We have added that in the background section.(line 50-57 p2)
Round 2
Reviewer 2 Report
no additional comments